# Evaluation of Low-Frequency Noise in MOSFETs Used as a Key Component in Semiconductor Memory Devices

Akinobu Teramoto 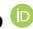

Research Institute for Nanodevice and Bio Systems, Hiroshima University, Higashi-Hiroshima 739-8527, Japan; teramo10@hiroshima-u.ac.jp; Tel.: +81-82-424-6266

**Abstract:** Methods for evaluating low-frequency noise, such as 1/f noise and random telegraph noise, and evaluation results are described. Variability and fluctuation are critical in miniaturized semiconductor devices because signal voltage must be reduced in such devices. Especially, the signal voltage in multi-bit memories must be small. One of the most serious issues in metal-oxide-semiconductor field-effect-transistors (MOSFETs) is low-frequency noise, which occurs when the signal current flows at the interface of different materials, such as $SiO_2/Si$. Variability of low-frequency noise increases with MOSFET shrinkage. To assess the effect of this noise on MOSFETs, we must first understand their characteristics statistically, and then, sufficient samples must be accurately evaluated in a short period. This study compares statistical evaluation methods of low-frequency noise to the trend of conventional evaluation methods, and this study's findings are presented.

**Keywords:** MOSFET; low-frequency noise; random telegraph noise; evaluation method; array test pattern





## 1. Introduction

Semiconductor devices have been basically progressed with the shrinking of MOSFETs (metal-oxide-semiconductor field-effect-transistors), which are used as the key component in them. The shrinkage has been performed following the rule of the constant electric field in MOSFETs, which decreases signal voltage [1,2]. In addition, power consumption of electronic devices has skyrocketed because the amount of digital data generated is growing at a rate faster than Moor's law [3,4]. The reduction of power consumption strongly requires decreasing supply voltage of MOSFETs because power consumption (P) is proportional to the square of the supply voltage ($V_{dd}$) as follows [5].

$$P = C_L V_{dd}^2 f \tag{1}$$

where $C_L$ represents the load capacitor and f represents the switching frequency of the circuit. The growth of clock frequency in the leading edge logic devices has stopped by exceeding the heat extraction capability. However, the downscaling has been continued to reduce the cost. In the other devices, the downscaling the device size has also been continued to reduce the power consumption and the other reasons. As a result, as the device size is reduced, the signal voltage of MOSFETs decreases. Memory devices' power consumption and supply voltage also have to be reduced [6–9] because of the same reasons as the logic devices and reducing a leakage current. On the other hand, a decrease in signal voltage degrades the reliability of electronic circuits, including analog and digital devices.

The logic (bit) error rate (LER) is given by the following equation.

$$LER = P_0 \int_{-\infty}^{\alpha} f_0(x)dx + P_1 \int_{\alpha}^{\infty} f_1(x)dx \tag{2}$$

where $P_0$ and $P_1$ represent the probabilities of signals "0" and "1", respectively, $\alpha$ represents the identification level between "0" and "1", and $f_0(x)$ and $f_1(x)$ represent noise amplitude

densities superimposed on "0" and "1", respectively. If $f_0$ and $f_1$ are Gaussian noise, the following equations apply.

$$f_0(x) = \frac{1}{\sqrt{2\pi\sigma^2}}e^{-\frac{x^2}{2\sigma^2}}, \; f_1(x) = \frac{1}{\sqrt{2\pi\sigma^2}}e^{-\frac{(x-A_S)^2}{2\sigma^2}} \tag{3}$$

where $A_S$ represents the signal amplitude and $\sigma$ represents the standard deviation of the noise. When $P_0 = P_1 = 1/2$ and $\alpha = A_S/2$ are assumed, the LER is given by the following equation from Equations (2) and (3).

$$\text{LER} = \frac{1}{2}\text{erfc}\left(\frac{A_S}{2\sqrt{2\sigma^2}}\right) \tag{4}$$

$$\text{erfc}(x) = 1 - \text{erf}(x) = 1 - \frac{2}{\sqrt{\pi}}\int_0^x e^{-t^2}dt = \frac{2}{\sqrt{\pi}}\int_x^\infty e^{-t^2}dt \tag{5}$$

When Nyquist transmission rate is the same as the signal band, the signal to noise (S/N) ratio (dB) is given by the following equation.

$$\frac{S}{N} = \frac{E_b}{N_0} = 20\log\left(\frac{A_S}{\sigma}\right) \tag{6}$$

where $E_b$ and $N_0$ represent signal and noise energy per second, respectively. From Equations (4) and (6), the LER is given by the S/N ratio as follows [10].

$$\text{LER} = \frac{1}{2}\text{erfc}\left\{\frac{1}{2\sqrt{2}}10^{\frac{1}{20}\left(\frac{S}{N}\right)}\right\} \tag{7}$$

Figure 1 shows the LER as a function of S/N (dB) and A/$\sigma$ ratios [10]. The LER decreases with an increase in S/N ($A_S/\sigma$) ratio. On the other hand, to guarantee that a system does not make a mistake even once during the operation period, the LER should be reduced as shown in the following equation.

$$\text{LER} \leq \frac{1}{N_L \times F \times T} \tag{8}$$

where $N_L$, F, and T represent the number of logic gates in a chip, the number of operations per second, and the guarantee period, respectively. For example, the LER should be less than $3 \times 10^{-26}$ for a circuit with $10^8$ logic gates, $10^9$ Hz operations, and a 10-year ($3 \times 10^8$ s) operation period, and then the S/N ($A_S/\sigma$) ratio should be greater than 26.5 dB (21.1).

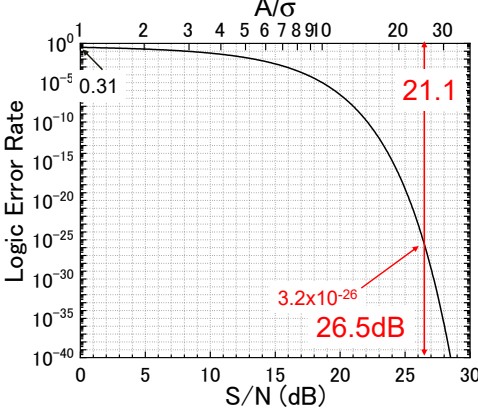

**Figure 1.** Logic error rate as a function of S/N and A/$\sigma$ ratios.

For example, we consider signal electrons required from a no error operation. Figure 2 shows (a) Variation in the number of electrons as a function of the number of signal electrons. (b) Logic error rate as a function of the number of signal electrons. When a signal is constructed by a constant number of electrons ($N_e$), the standard deviation of the number of electrons is $(N_e)^{1/2}$, and then the A/σ ratio is $(N_e)^{1/2}$. As a result, the number of electrons must be greater than 445 to maintain an S/N ($A_S/\sigma$) ratio of 26.5 dB (21.1). If the number of signal electrons is 1, the LER must be equal to 0.3. Then, such a system produces the wrong output once every three calculations.

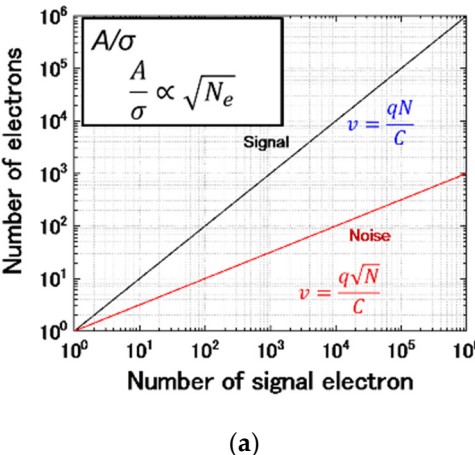

(**a**)

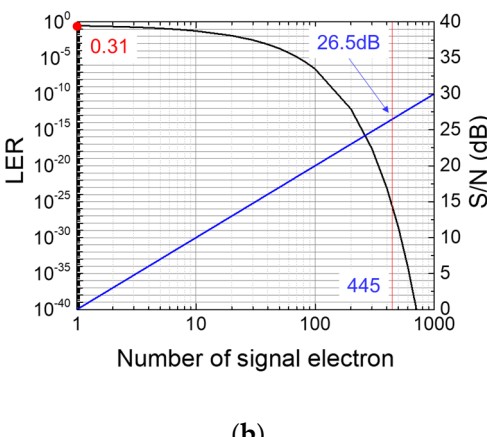

(**b**)

**Figure 2.** (**a**) Variation in the number of electrons as a function of the number of signal electrons. (**b**) Logic error rate as a function of the number of signal electrons.

The electronic circuit will be influenced by noise, such as thermal noise, quantum noise, and flicker noise. The noise voltage ($v_{nf}$) in $1/f$ noise is defined by the following equation.

$$v_{nf} = \sqrt{\frac{K_F}{C_{OX} \times L \times W} \ln\left(\frac{f_H}{f_L}\right)} \tag{9}$$

where $K_F$ represents the flicker noise coefficient, $f_L \sim f_H$ is the frequency period for device operation, $C_{OX}$, L, and W represent the gate oxide capacitance, gate length, and gate width of a MOSFET, respectively. Noise increases with device shrinkage because $v_{nf}$ is inversely proportional to $\sqrt{C_{OX}LW}$ [11–14]. It has been pointed out that $1/f$ noise may influence not only analog devices, but also digital devices when device shrinkage and the decreasing signal voltage are moved on [15]. Random telegraph noise (RTN), another low-frequency noise also affects electronic devices, such as CMOS image sensor [16–21], static random access memory (SRAM) [22–25], dynamic random access memory (DRAM) [25], and flash memory [26–32]. Low-frequency noise, such as $1/f$ noise and RTN, have high variability [33,34] because they must be statistical phenomena by nature, and statistical analysis is required to fully understand this phenomenon. The conventional evaluations of the noise in MOSFETs have performed with a few sample numbers, and then we could understand only typical noise characteristics of MOSFETs having relatively large noise. However, we need statistical information of the noise for the design of LSI. Then, low-frequency noise statistical evaluation methods and evaluation results are described in this study.

## 2. Evaluation Methods

### 2.1. Test Pattern for Noise Evaluation

The test structure is constructed using 0.22 μm, 1-poly 2-metal standard CMOS technology and includes n-MOSFETs of various gate sizes [35–38] as shown in Table 1. The measured MOSFETs are arrayed in 1024 rows and 1776 columns (total number of MOSFETs:

1217856) in a chip at 5 μm intervals. The size of the MOSFETs, and their number, and location in a chip are shown in Table 1. The test chip has an area of 5.5 mm × 14 mm. The gate insulator is formed by pyrogenic oxidation and is 5.8 nm thick.

**Table 1.** Number of n-MOSFETs for each transistor size.

| Gate Length (μm) | Gate Width (μm) | Number of MOSFETs | Supply Voltage (V) |
|---|---|---|---|
| 0.22 | 0.28 | 131,072 (128 × 1024) | |
| 0.22 | 0.30 | 131,072 | |
| 0.24 | 0.30 | 131,072 | |
| 0.24 | 1.5 | 131,072 | |
| 0.24 | 15 | 131,072 | |
| 0.4 | 1.5 | 32,768 (128 × 256) | |
| 0.4 | 15 | 32,768 | 2.5 |
| 1.2 | 0.3 | 65,536 (64 × 1024) | |
| 1.2 | 1.5 | 65,536 | |
| 4.0 | 0.30 | 65,536 | |
| 4.0 | 1.5 | 65,536 | |
| 0.24 | 0.30 | 32,768 (AR:100) | |
| 0.24 | 0.30 | 4096 (16 × 256, AR:1000) | |
| 0.24 | 0.30 | 1344 (32 × 42, AR:10000) | |
| 0.4 | 1.5 | 131,072 | |
| 0.4 | 15 | 32,768 | |
| 1.2 | 15 | 16,384 (64 × 256) | 3.3 |
| 4.0 | 15 | 16,384 | |

AR: Antenna ratio.

A schematic block diagram of a test pattern is shown in Figure 3a [35,37,39,40]. This is composed of MOSFETs measured in arrayed unit cells, vertical and horizontal shift registers for addressing measured MOSFETs, MOSFETs located on each column for current control of measured MOSFET, analog memories for storing the source voltage of the measured MOSFETs within one line, and a source follower circuit for amplifying the output signal. The drain ($V_D$) and gate ($V_G$) voltage in measured MOSFETs and the gate voltage applied to current source MOSFETs ($V_{REF}$) are supplied from the external voltage source simultaneously. $V_{DD}$ and $V_{SS}$ are the supply voltages in the peripheral circuits and ground voltage, respectively. The measured MOSFET and current source transistor construct a source follower circuit using a select transistor. This test structure uses simple peripheral circuits. Therefore, it can be used to evaluate various MOSFETs with varying gate lengths, gate widths, gate insulator films, thicknesses, and other characteristics. Figure 3b shows the circuit schematic of a unit cell and current source transistor in Figure 1, which is the principle of this measurement. A unit cell is constructed with a measured MOSFET and a select transistor. When the current source transistor operates at a saturation region, $I_{REF}$ is independent of the voltage between the source and drain in the current source transistor ($V_{out}$). When the gate bias of the select transistor ($\Phi_x$) is applied from a vertical shift register, $I_{REF}$ flows into the measured MOSFET. The output voltage ($V_{out}$) is indicated as follows.

$$V_{gs} = V_G - V_{out} - I_{REF} \cdot R_{select} \approx V_G - V_{out} \qquad (10)$$

where $R_{select}$ is the channel resistance of the select switch transistor. The select transistor must be operated in the linear region to have sufficient high channel conductivity compared with the measured MOSFET, and then, $I_{REF} \cdot R_{select}$ can be neglected. The output signal can be obtained as a source voltage for each measured MOSFET by shift register scanning, and then 1.2 million MOSFETs can be measured within approximately 0.7 s. The electrical characteristics of the measured MOSFETs can be observed as the $V_{gs}$ included in the output voltage $V_{out}$ (Figure 3b). In this frame measurement mode, each MOSFET can be measured every 0.7 s. This test pattern has another measurement mode, which can measure a specific MOSFET every 1 μs.

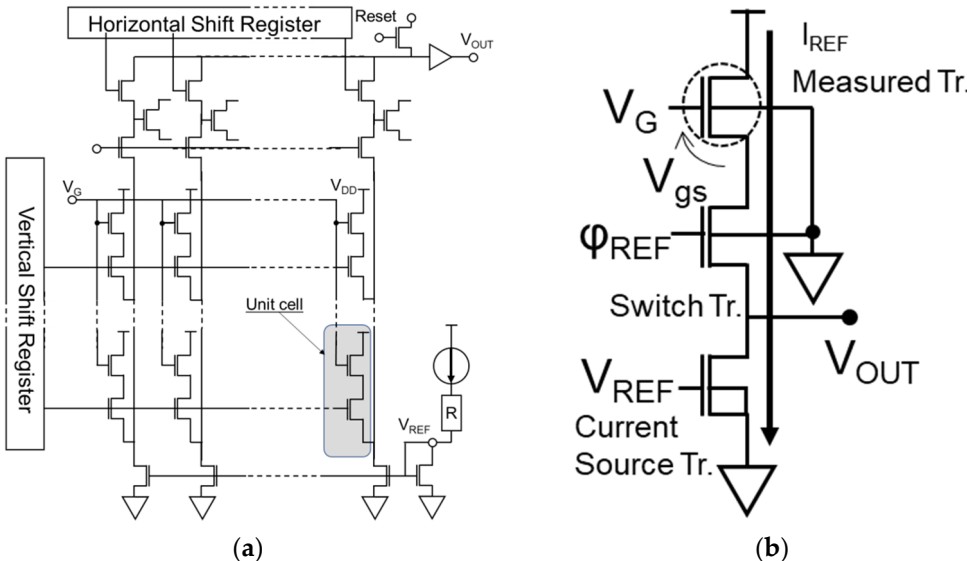

**Figure 3.** (**a**) Schematic block diagram of a test pattern. (**b**) Unit cell.

### 2.2. Extraction of Amplitude and Time Constant of RTN

Two-level type RTN is characterized by only three parameters, which are the mean time to capture ($<\tau_c>$), mean time to emission ($<\tau_e>$), and amplitude ($\Delta V_{gs}$). The time constants correspond to two physical states of a trap, that is, $\tau_c$ and $\tau_e$ represent spans in a low $V_{gs}$ level (carrier trapping state) and high $V_{gs}$ level (carrier emission state), respectively (Figure 4a). The RTN amplitude $\Delta V_{gs}$ is defined as the difference between two normal distributions in a voltage histogram (Figure 4b). We extract the time constants by fitting the distributions of $\tau_c$ and $\tau_e$ to the exponential distribution ($Ae^{-t/<\tau>}$) because the phenomenon is governed by the Poisson process. The time constants can be extracted with μs accuracy using the specific MOSFET measurement mode.

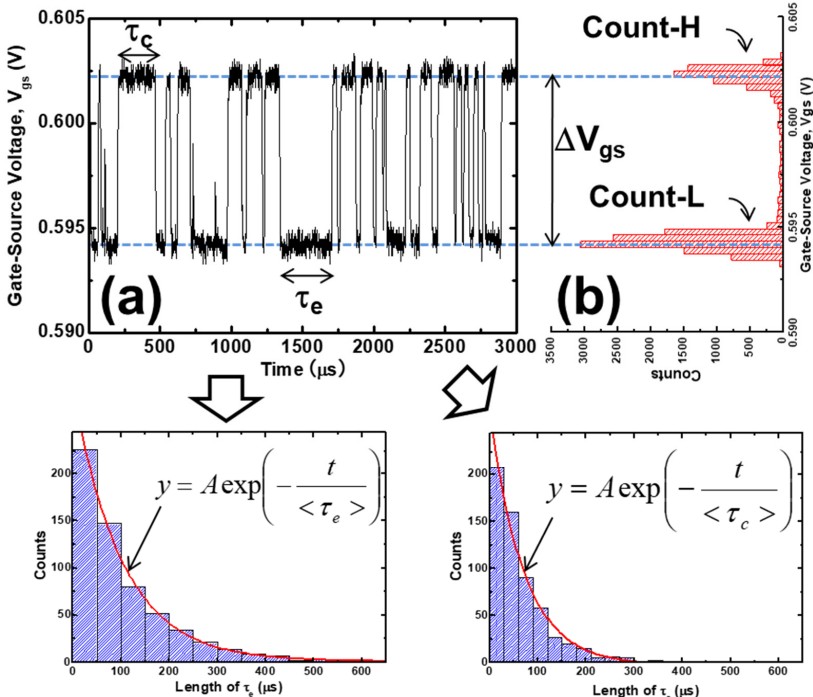

**Figure 4.** Definitions and extractions of two time constants ($<\tau_c>$, $<\tau_e>$) and amplitude ($\Delta V_{gs}$) from RTN (**a**)waveform data, (**b**)histogram of Vgs.

The time constant ratio $<\tau_e>/<\tau_c>$ is also an important parameter in RTN because the energy level of a trap that causes RTN is related to the constant ratio as follows [12,41–43].

$$\frac{<\tau_c>}{<\tau_e>} = g\exp\left(\frac{E_T - E_F}{kT}\right) \tag{11}$$

where $E_T$ and $E_F$ represent the energy of the trap and Fermi energy of the channel, respectively; k, T, and g represent Boltzmann constant, temperature, and degeneracy factor, respectively, where g is assumed to 1. Then, the energy of the trap level is indicated by (12).

$$E_T - E_F = kT\ln\left(\frac{<\tau_c>}{<\tau_e>}\right) \tag{12}$$

We can use the frame measurement mode to extract the time constant ratio, and the 1.2 million MOSFETs can be measured 10,000 times in 7000 s (sampling period = 0.7 s). An average of the time constant ratio $<\tau_e>/<\tau_c>$ is the same as Count-L/Count-H (shown in Figure 4b), where Count-L and count-H are the numbers of low and high states, respectively [41–44]. When the time constant is greater than a sampling frequency of 0.7 s, the detected number of transition times is the same as the transition time of RTS characteristics. However, when the time constant is less than 0.7 s, the detected number of transition times is less than the real one; however, it is proportional to the real one because the absolute value of the time constant, which is less than the sampling frequency, cannot be extracted in this measurement. Then, the number of transition times is defined as the detected ones in the sampling frequency of 0.7 s.

### 2.3. Root Mean Square of RTN Waveform

The root mean square (RMS) of the signal waveform is often used for the representative parameter of noise [45], and the RMS of the output voltage $V_{RMS}$ is defined as follows in this study.

$$V_{RMS} = \sqrt{\frac{\sum_{i=1}^{N}\left(V_{out,i} - \overline{V_{out}}\right)^2}{N-1}} = A\frac{\sqrt{<\tau_e><\tau_c>}}{<\tau_e> + <\tau_c>} \tag{13}$$

where $V_{out,i}$, $\overline{V_{out}}$, N, and A are the output voltage at ith sampling, average of $V_{out}$, sampling numbers, and the amplitude of two-state RTN, respectively. Using $V_{RMS}$, we can obtain MOSFETs with high noise from many measured MOSFETs. Figure 5 shows the relationship between $V_{RMS}$ and RTN waveform. The waveform with large RTN corresponds to large $V_{RMS}$.

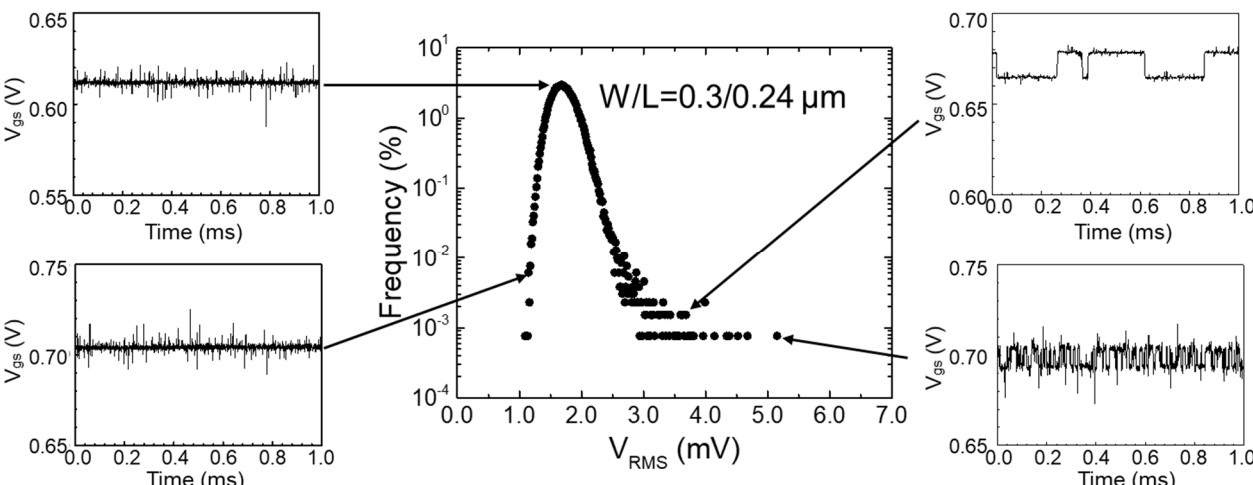

**Figure 5.** Relation between $V_{RMS}$ and RTN waveform.

## 3. Results and Discussion

### 3.1. Statistical Evaluation of RTN Characteristics

The 1/f noise increases with downscaling of MOSFETs, as mentioned above, and RTN also increase with the downscaling [12,36,37]. Figure 6 shows the Gumbel plot of $V_{RMS}$ for the various MOSFET sizes [36,37]. A large $V_{RMS}$ can be observed in small-size MOSFETs (L/W = 0.22/0.28, 0.22/0.3, 0.24/0.3 μm). In this experiment, noise cannot be observed in large MOSFETs because the floor noise is relatively high at ~2.5 mV.

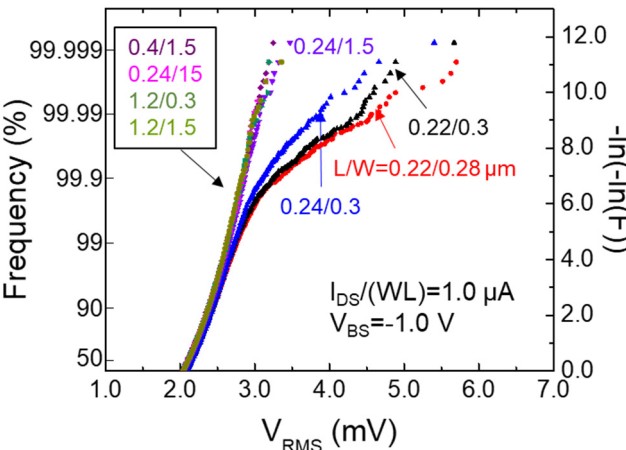

**Figure 6.** Gumbel plot of $V_{RMS}$ for various MOSFET sizes. The measured sizes of MOSFETs are L/W = 0.22/0.28, 0.22/0.3, 0.24/0.3, 0.24/1.5, 0.4/1.5, 0.24/15, 1.2/0.3, 1.2/1.5 μm.

Figure 7a shows the Gumbel plot of $V_{RMS}$ for the various $I_{DS}$ varied from 0.13 to 12.7 μA. The sizes of MOSFETs are L/W = 0.22/0.28 μm and $V_{BS} = -1.0$ V [39]. The data in (a) and (b) were measured by the frame and specific MOSFET measurement modes, respectively. The number of MOSFETs with large noise increases with decreasing $I_{DS}$, which is controlled by $V_{gs}$. This means that the event probability of large noise increases with decreasing $V_{gs}$ because the number of channel electrons decreases with decreasing $V_{gs}$, and then the effect of a trapped electron charge becomes large with decreasing number of channel electrons. The number of channel electrons also decreases with the shrinkage of transistor size shown in Figure 6, and then, the probability increases with decreasing channel size. Figure 7b shows the waveform of typical MOSFETs for $I_{DS}$ of 0.13, 0.38, and 1.3 μA [39]. The time constants and amplitude are modulated by $I_{DS}$. With increasing $I_{DS}$ ($V_{gs}$), amplitude and $\tau_c$ decrease, whereas $\tau_e$ slightly increases. An increase in $V_{gs}$ decreases $E_T$-$E_F$, and then, the time to capture decreases as shown in Equation (11). The difference between the modulation of $\tau_e$ and $\tau_c$ is discussed later. The modulation of amplitude is caused by a decrease in the number of electrons, as discussed above. It is considered that decreasing the time to capture and increasing amplitude with decreasing $V_{gs}$ increases the event probability of large noise. Figure 8a shows the Gumbel plot of $V_{RMS}$ for the various back bias ($V_{BS}$). $V_{BS}$ varied from $-0.075$ to $-1.38$ V, and Figure 8b shows the waveform of typical MOSFETs for $V_{BS}$ of 0.6 1.0 and 1.3 V [39]. The probability increases with the absolute value of $V_{BS}$ in (a). In this experiment, $I_{DS}$ was constant at 1.0 μA, and this means that the number of electrons was almost the same for each $V_{BS}$. Increasing $V_{BS}$ caused channel percolation [46–49], making the channel thickness narrow and percolated and increasing electron energy [50]. The probability is increased by channel percolation [46,47], and the varying electron energy modulates the time constants. In MOSFETs with RTN, the amplitude does not increase with increasing $V_{BS}$ because the number of electrons is the same for each $V_{BS}$. This means that channel percolation increases the probability of RTN generation.

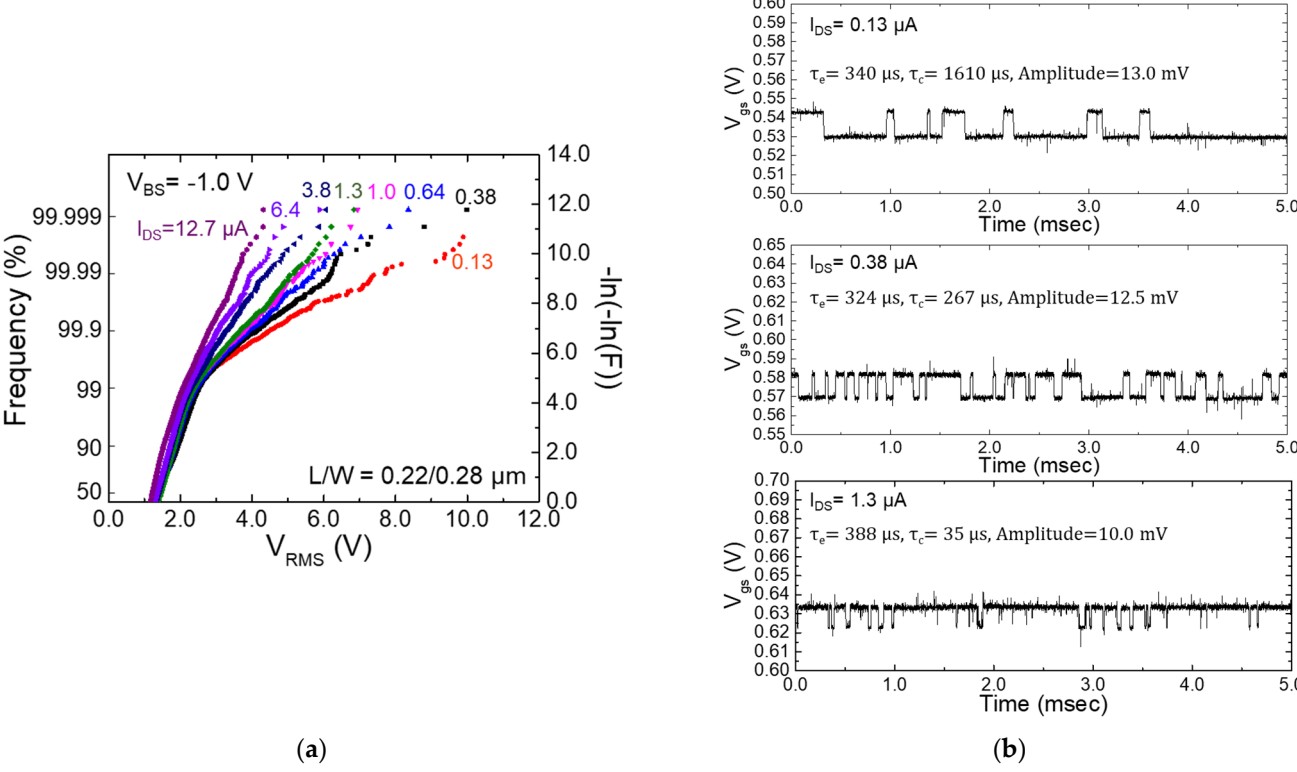

**Figure 7.** (**a**) Gumbel plot of $V_{RMS}$ for the various $I_{DS}$. $I_{DS}$ varied from 0.13 to 12.7 μA, and (**b**) waveform of typical MOSFETs for $I_{DS}$ of 0.13, 0.38, and 1.3 μA. The sizes of MOSFETs are L/W = 0.22/0.28 μm and $V_{BS}$ = 1.0 V.

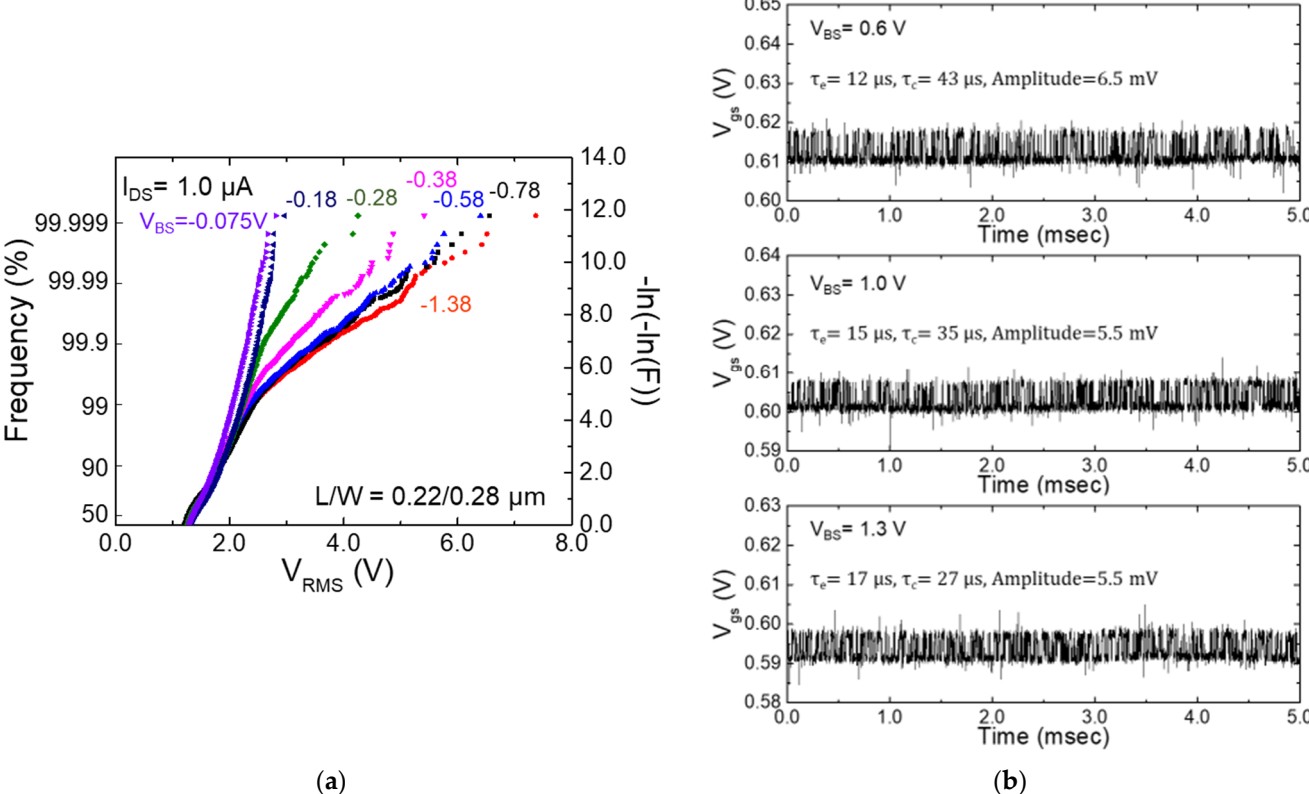

**Figure 8.** (**a**) Gumbel plot of $V_{RMS}$ for various $V_{BS}$. $V_{BS}$ varied from 0.075 to 1.38 V, and (**b**) waveform of typical MOSFETs for $V_{BS}$ of 0.6 1.0 and 1.3 V. The sizes of MOSFETs are L/W = 0.22/0.28 μm and $I_{DS}$ = 1.0 μA.

Figure 9 shows the Gumbel plot of the RTN amplitude for MOSFETs with varying channel doping [51]. The channel percolation is accelerated by increasing channel doping concentration [46–48]. This figure shows that the probability of the number of MOSFETs with large amplitude increases with doping concentration. RTN is increased by channel doping as well as doping the concentration near the source and drain regions. Figure 10 shows the Gumbel plot of $V_{RMS}$ for various Halo implantation concentrations [52]. The number of MOSFETs with large RTN increases with an increase in Halo implantation concentration. This indicates that the high dose in the channel region or near the source/drain region results in high RTN because of channel percolation enhancement.

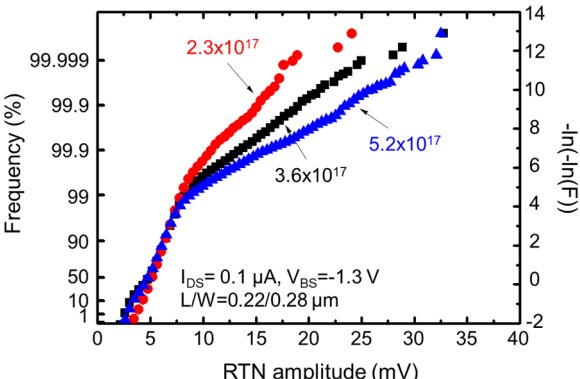

**Figure 9.** Gumbel plot of the RTN amplitude for MOSFETs with varying channel doping. The channel doping concentrations are varied $2.3 \times 10^{17}$, $3.6 \times 10^{17}$, $5.2 \times 10^{17}$ cm$^{-3}$, respectively.

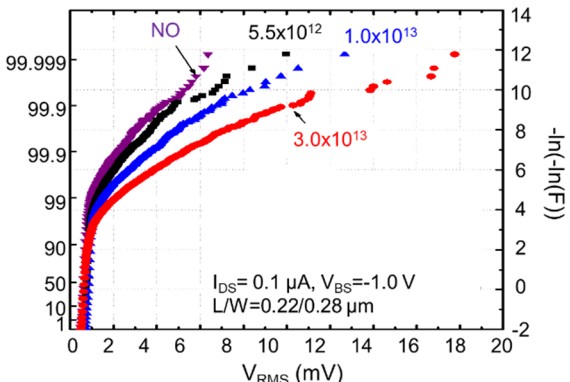

**Figure 10.** Gumbel plot of the RTN amplitude for MOSFETs with varying channel doping. The channel doping concentrations are varied $2.3 \times 10^{17}$, $3.6 \times 10^{17}$, $5.2 \times 10^{17}$ cm$^{-2}$, respectively.

Figure 11 shows the energy band diagrams and energy distribution of traps causing RTN for $V_{gs}$ of 0.57, 0.53, and 0.46 V, respectively [41,42]. The difference between $E_T$ and $E_F$ for electrons is calculated using Equation (11). The blue shading and red solid bars in Figure 11 show the energy distribution of traps causing RTN in each measurement condition and common traps in all conditions, respectively. Although the shape of the distribution of each $V_{gs}$ is almost the same, and the energy of common traps in all $V_{gs}$ increases with decreasing $V_{gs}$. The conduction band edge ($E_C$), the bottom sub-band energy ($E_{sub}$), and 2nd sub-band energy ($E_{2nd}$) in the inversion layer and $E_F$ are indicated in Figure 11 [50]. The energy levels of sub-bands were calculated using Equation (14) [50].

$$E_j = \left[ \frac{3hqE_s}{4\sqrt{2m_x}} \left( j + \frac{3}{4} \right) \right]^{\frac{2}{3}}, \ j = 0, 1, \tag{14}$$

where $E_s$ is the electric field, h and $m_x$ represent Planck's constant and effective mass of electrons, respectively. $E_j$ is jth sub-band energy, and $E_{sub}$ and $E_{2nd}$ represent $E_0$ and $E_1$,

respectively. The main energy distribution for each $V_{gs}$ locates higher energy than the conduction band edge. It is considered that the energy level of traps is widely distributed, and the energy of the detected traps is determined by the electron energy in $E_{sub}$ and $E_{2nd}$. Conversely, the energy of common traps in all $V_{gs}$ increases with decreasing $V_{gs}$ because the influence of trap energy on $V_{gs}$ is larger than that of electron energy in the channel.

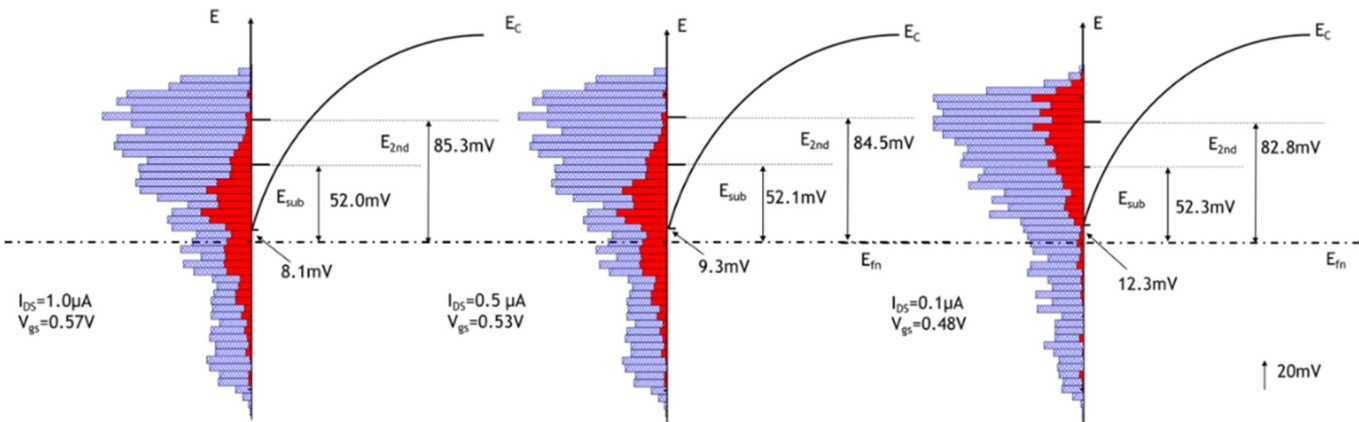

**Figure 11.** Conduction band diagrams and energy distribution of the traps causing RTN for $V_{gs}$ of 0.57, 0.53, and 0.46 V, respectively.

### 3.2. Multi-State RTN

Large $V_{RMS}$ RTN includes both two-state and multi-state RTN [34,53–56], which is considered to be generated by multi-traps. The analysis of trap characteristics, such as time constants and amplitude, in multi-state RTN is more difficult than that of two-state. Figure 12 shows the appearance probability of RTN with two, three, four, and more than four states. A large RTN ($V_{RMS} > 680\ \mu V$) was obtained by a frame measurement mode of the sampling period of 0.7 s/frame in $I_{DS} = 1\ \mu A$. 131,072 MOSFETs (L/W = 0.22/0.28 µm) were measured, and 2575 MOSFETs with large $V_{RMS}$ can be extracted. Then, we selected MOSFETs with large RTN and measured them by a specific measurement mode of a sampling period of 1 µs and a long sampling time of 10 min (sampling points = $6 \times 10^8$) for the same bias condition [57,58]. Figures 13–15 show the (a) waveform, (b) time lag plot (TLP), and (c) histogram for typical three-, four-, and six-state RTN. The number of peaks and the transition of each state can be understood via TLP [53,54,56].

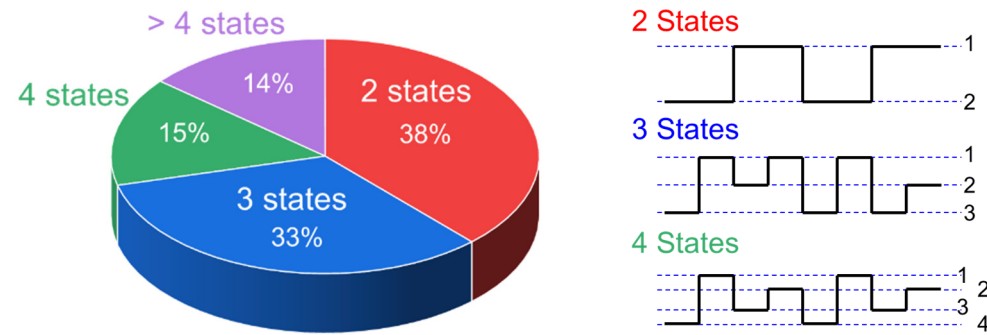

**Figure 12.** Appearance probability of RTN with two, three, four, and more than four states. 2575 MOS-FETs with large $V_{RMS}$ can be extracted from 131,072 MOSFETs (L/W = 0.22/0.28 µm).

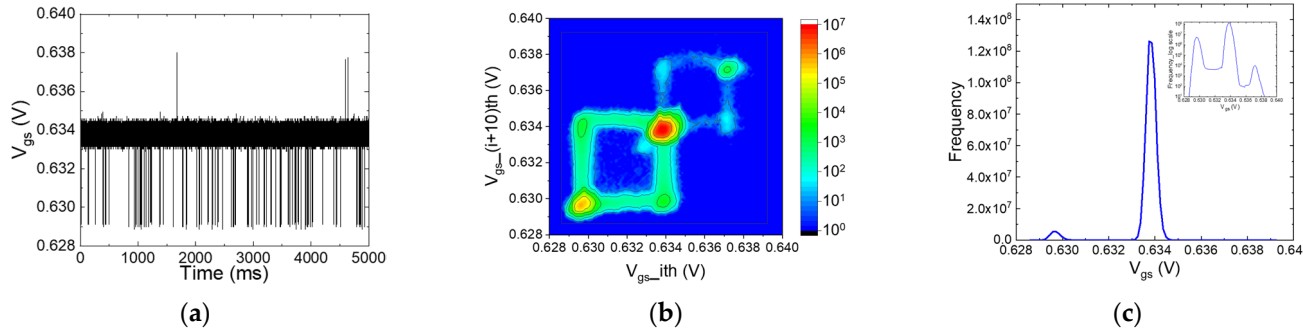

**Figure 13.** (**a**) Waveform, (**b**) TLP, and (**c**) histogram for a typical three-state RTN. The inset figure (**c**) shows the same data of the vertical axis of the log (histogram).

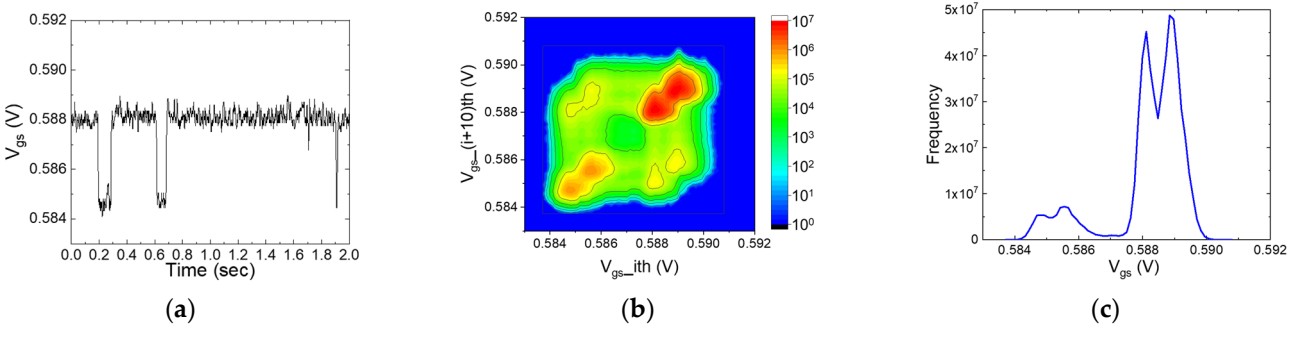

**Figure 14.** (**a**) Waveform, (**b**) TLP, and (**c**) histogram for a typical four-state RTN.

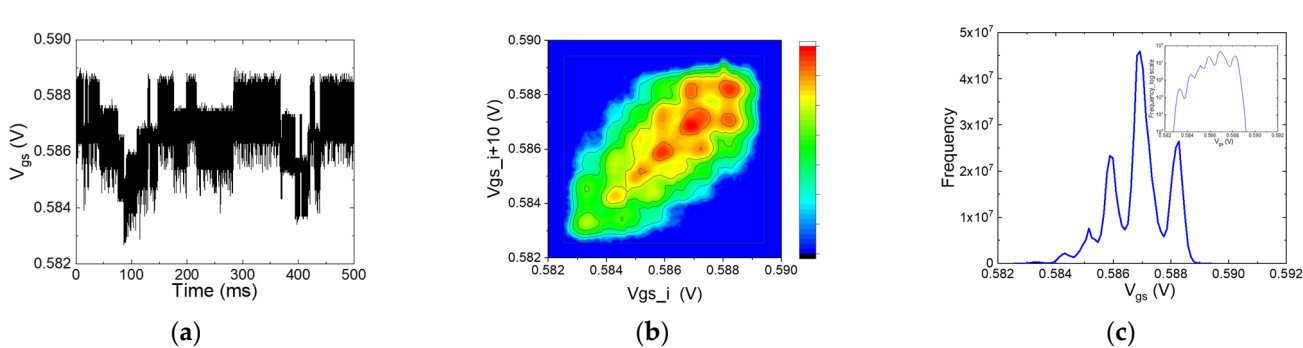

**Figure 15.** (**a**) Waveform, (**b**) TLP, and (**c**) histogram for a typical six-state RTN. The insect figure in (**c**) shows the same data of the vertical axis of the log (histogram).

Figures 13–15 show the (a) waveform, (b) time lag plot (TLP), and (c) histogram for typical three-, four-, and six-state RTN. The number of peaks and the transition of each state can be understood via TLP [53,54,56]. Figures 13, 14 and 15b show the relationship of ith and (i + 10)th $V_{gs}$ for the constant IDS. As shown in Figure 13b, transitions occur not from the lowest state to the highest state, but only via the medium state. When the trapping probability of some traps is even, the number of states should be even, and the transition from one position to the next can occur. To begin with, an odd number of states implies that the trapping probability for each trap is not independent of each other. A similar transition phenomenon occurs even in a four-state case. As shown in Figure 14b, transitions did not occur from the lowest state to the highest state or from the second-lowest state to the second-highest state. This also means that there are more than two traps, and the probability of trapping for each trap is not independent of each other. The characteristics of multi-trap RTN can be understood via TLP and waveforms [56]; however, these analyses become more difficult as the number of states increases.

### 3.3. Time Constants in Individual RTN

As the extraction of multi-trap phenomena is difficult, as discussed in Section 3.2, we discuss the time constants and amplitude only in two-state RTN [59]. Figure 16 shows (a) $\tau_c$ and (b) $\tau_e$ as a function of $I_{DS}$, respectively. These data are measured at $I_{DS}$ of 0.1, 0.3, 1.0, 3.0, and 5.0 µA. The data in Figure 16 show how parameters from all two-level RTN can be extracted in common under four or five $I_{DS}$, and thus, the number of selected data points was 22.

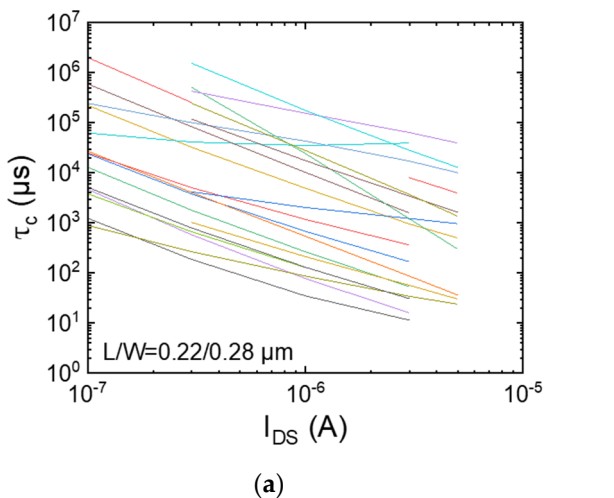
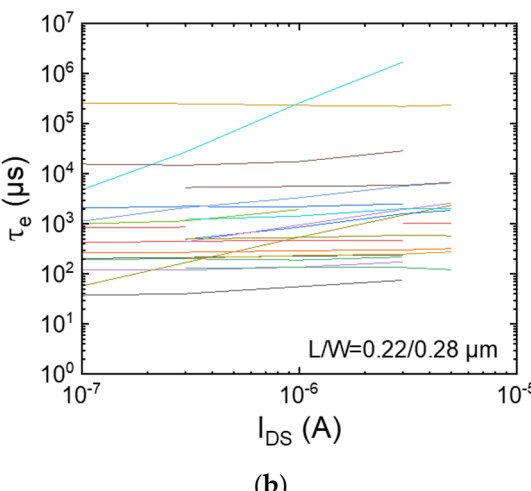

(**a**)                                                        (**b**)

**Figure 16.** (**a**) $\tau_c$ and (**b**) $\tau_e$ as a function of $I_{DS}$. These data are measured at $I_{DS}$ of 0.1, 0.3, 1.0, 3.0, and 5.0 µA. The number of data points is 22.

Figure 17a shows $\tau_c/\tau_e$ and $E_T-E_F$ as a function of $I_{DS}$ and Figure 17b band diagram of MOS structure for changing $V_{GS}$ ($I_{DS}$). $\tau_c/\tau_e$ and $E_T-E_F$ are calculated from the data in Figure 16 and Equation (12), respectively. $\tau_c$ and $\tau_e$ decrease and increase with an increase in $I_{DS}$ ($V_{GS}$), and the absolute slope of $\tau_c$ is significantly larger than that of $\tau_e$. Large trap energy ($E_T$) decreases with an increase in $V_{GS}$ than that of channel electron ($E_C$: bottom energy of conduction band). Then, with increasing $V_{GS}$ ($I_{DS}$), the energy barrier from the channel electron to the trap decreases and that from the trapped electron to the channel increases. As a result, the time to capture and time to emission decreases and increases, respectively, with increasing $V_{GS}$ ($I_{DS}$). The transition probability depends on the energy barrier height between a trap and channel. $\tau_e$ depends only on the energy barrier because only one electron is captured in a trap. Meanwhile, $\tau_c$ depends not only on the energy barrier, but also on the number of electrons in a channel because the number of channel electrons increases as $V_{GS}$ ($I_{DS}$) increases. Then, the dependency of $\tau_c$ on $I_{DS}$ is more significant than that of $\tau_e$. $E_T-E_F$ in Figure 17 changed by approximately 175 mV during $I_{DS}$ ($V_{GS}$) from 0.1 (0.53V) to 5.0 µA (0.75V). Based on these values, $E_T-E_F$ changes by 0.18 V, whereas $V_{GS}$ changes by 0.22 V. The distance between the traps and the channel was 4.6 nm due to the gate oxide thickness of 5.7 nm. However, $\tau_c$ depends not only on the trap energy, but also on the number of channel electrons; thus, $E_T-E_F$ values cannot be calculated using Equation (12). The distance is considered to be shorter than the calculated value. $\tau_e$ values for almost all samples monotonically increased with increasing $I_{DS}$. This suggests that the distance from the trap to the channel is shorter than that to the gate electrode. The distance between the trap and channel is shorter than 2.85 nm, which is the center of the gate oxide thickness. Figure 18a,b show the amplitude and transition frequency as a function of $I_{DS}$ for the same samples as those in Figures 16 and 17, respectively. For a sufficiently long measuring period, the transition frequency (TF) was calculated using the following equation.

$$\text{TF} \approx \frac{N_e}{\tau_e} \approx \frac{N_c}{\tau_c} \tag{15}$$

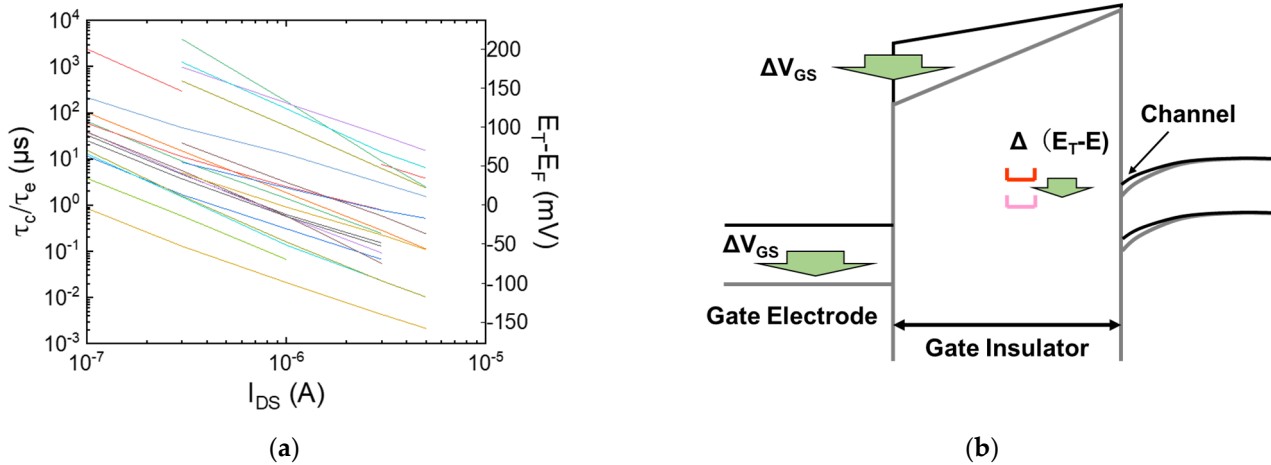

**Figure 17.** (**a**) shows $\tau_c/\tau_e$ and $E_T$-$E_F$ as a function of $I_{DS}$, and (**b**) band diagram of MOS structure for varying $V_{GS}$ ($I_{DS}$). $\tau_c/\tau_e$ and $E_T$-$E_F$ are calculated from the data in Figure 16 and Equation (12), respectively.

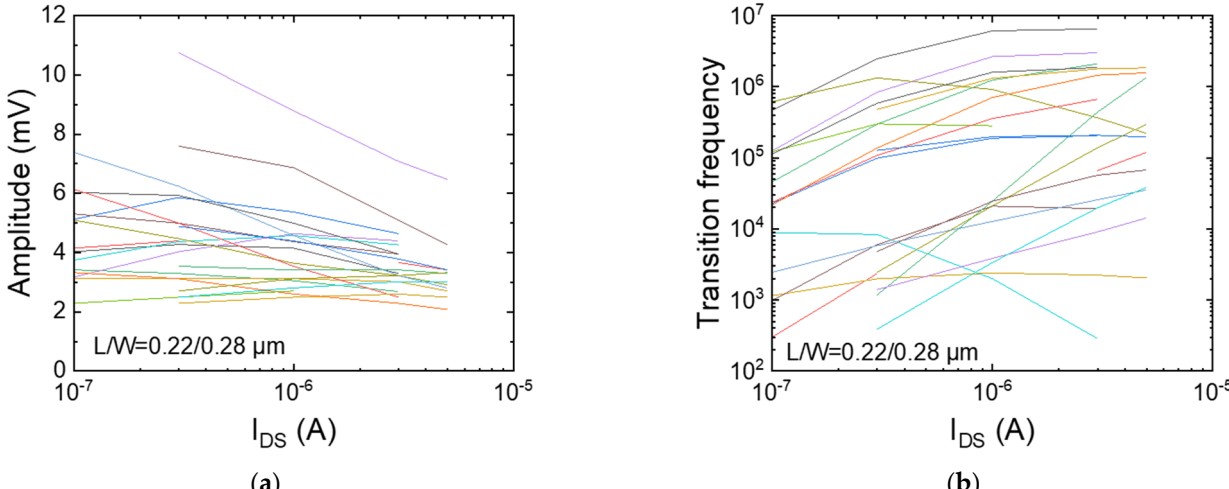

**Figure 18.** (**a**) Amplitude and (**b**) transition frequency as a function of $I_{DS}$.

The amplitude decreases and the TF increases with an increase in $I_{DS}$ for almost all samples. This is caused by the increase in the number of channel electrons as $I_{DS}$ increases. However, the amplitude and TF of some samples did not exhibit monotony, which is due to the percolation channel effect [46–49]. The distance between the channel and trap changes as $I_{DS}$ ($V_{GS}$) changes because of the formation of the percolation channel.

### 3.4. Effect of Drain Current on Appearance Probability and Amplitude

Figure 19 shows the Gumbel plot of the $V_{RMS}$ for 18,048 MOSFETs. $I_{DS}$ varied from 0.1 to 20 μA. The floor noise in this experiment was smaller than the others and was approximately 35 μ$V_{RMS}$ [60]. In Figure 7, $V_{RMS}$ decreases with an increase in $I_{DS}$ for all $V_{RMS}$. In Figure 19, larger $V_{RMS}$ can also be observed in small $I_{DS}$ in relatively large $V_{RMS}$ regions. However, the higher appearance probability in large $I_{DS}$ than that in small $I_{DS}$ for the small $V_{RMS}$ region of less than 500 μV could not be observed in Figure 7 because the floor noise was approximately 1 mV in that experiment. The amplitude characteristics are the same as the $V_{RMS}$ characteristics, and the distribution of the time constants is the same for all conditions [60]. Figure 20 shows the frequency of RTN with two, three, and more than three states in 18,048 MOSFETs. The frequency of all states increases with $I_{DS}$.

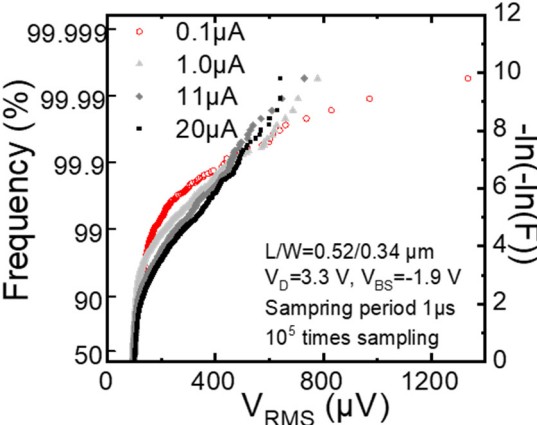

**Figure 19.** Gumbel plot of the $V_{RMS}$ for 18,048 MOSFETs. $I_{DS}$ varied from 0.1 to 20 μA. The floor noise was a 35-μ$V_{RMS}$.

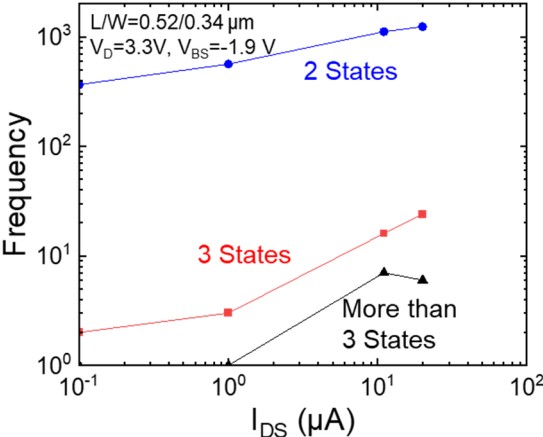

**Figure 20.** Frequency of RTN with two, three, and more than three states in 18,048 MOSFETs.

It is assumed that the probability that electrons and percolation paths are close to each other increases as $I_{DS}$ increases, increasing the number of electrons in the channel and the number of percolation paths. Figure 21 shows the Gumbel plot for the amplitude of two-state RTN. Notably, frequency sometimes increases with an increase in $I_{DS}$ even though the effect of trapped electrons on the channel decreases with an increase in the electron density in the channel.

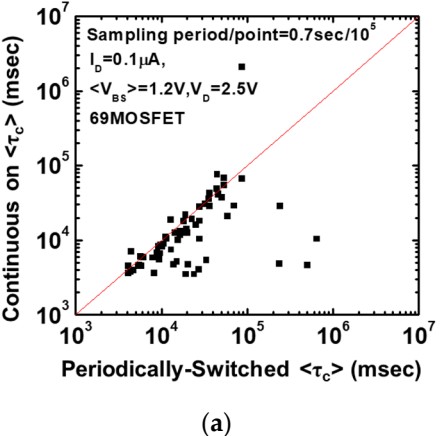

(a)

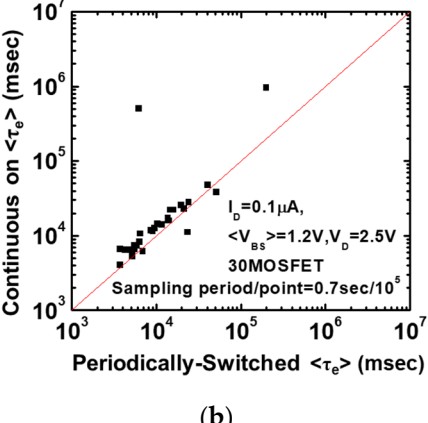

(b)

**Figure 21.** Relationship of time constants between continuous on-state and periodically switched conditions. (**a**) $\tau_c$, (**b**) $\tau_e$.

### 3.5. Modulation of Time Constants

In Section 3.3, the time constants were measured under constant conditions. The differences between time constants and VRMS in continuous on-state and periodically switched conditions are discussed in this section [61]. As shown in Figure 16, $V_{GS}$ dependance on $\tau_c$ is larger than that of $\tau_e$, and this suggests that the cycle period ($\tau_c + \tau_e$) in on-state is longer than that in off-state because the $V_{GS}$ of off-state is smaller than that of on-state. Figure 21a,b show the time constant relationship between continuous on-state and periodically switched conditions. In the periodically switched condition, MOSFETs cycled for 10.6 msec in 700 ms, which was a measurement cycle. Although $\tau_e$ of almost all samples are the same in both conditions, $\tau_c$ of some samples in the periodically switched condition is larger than that in continuous on-state. Figure 22 shows (a) histogram of the $V_{RMS}$ difference between continuous on-state and periodically off-state ($\Delta V_{RMS}$) and (b) schematic waveform of RTN in continuous on-state and periodically off-state, respectively. Though $V_{RMS}$ of 5% samples was increased, that of 95% was not changed or decreased in the periodically switched condition. As a result, $V_{RMS}$ decreased in the periodically switched condition, with a few exceptions. This suggests that $V_{RMS}$ can be reduced by the modulation of operation conditions, even in the same MOSFET.

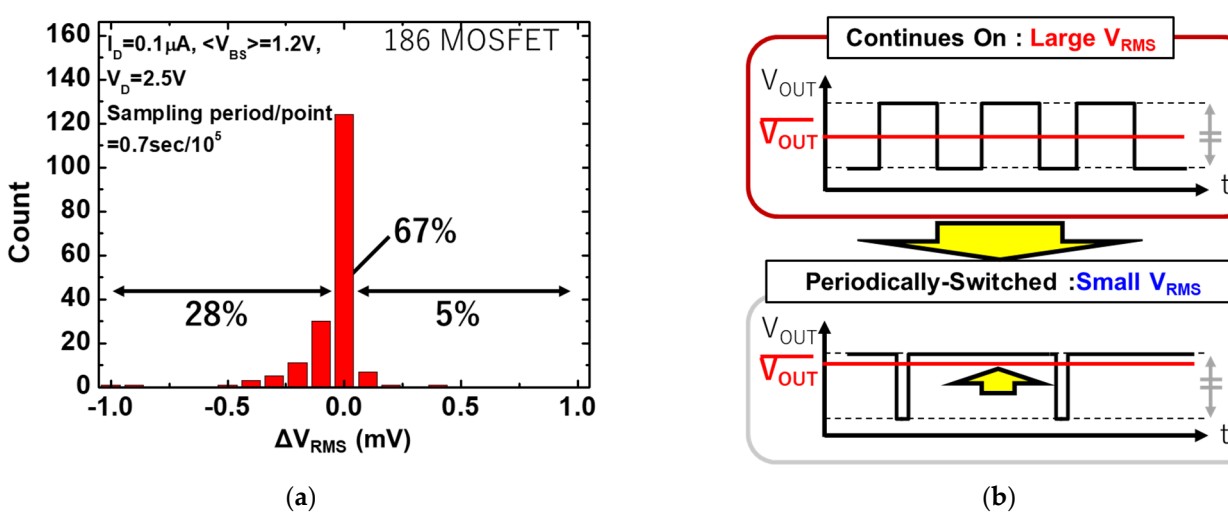

|   |   |
|---|---|
| (a) | (b) |

**Figure 22.** (**a**) Histogram of the $V_{RMS}$ difference between continuous on-state and periodically off-state ($\Delta V_{RMS}$), and (**b**) schematic waveform of RTN in continuous on-state and periodically off-state.

### 3.6. Device Structure Dependence of RTN

In the above results and discussions in Sections 3.1–3.5, the dependance of RTN characteristics on operation conditions is mainly described. In Section 3.6, the dependance of RTN characteristics on the structure of MOSFETs, such as buried channel MOSFETs and asymmetric source–drain structure MOSFETs, are described [57,62–65].

#### 3.6.1. Buried Channel MOSFETs

Figure 23 shows the structure of buried channel MOSFETs studied in this work. To discuss the effects of n-Si layer widths and the distance between the channel and $SiO_2$/Si interface, n-Si layer width was varied to be 0, 10, 25, and 60 nm for standard, narrow, middle, and deep samples formed by arsenic ion implantation, respectively, and the high-energy ion implantation created not only a deep channel, but also a wide channel in the vertical direction to the $SiO_2$/Si interface. Figures 24 and 25 show the Gumbel plots of $V_{RMS}$ for the standard, narrow, middle, and deep samples and the $V_{BS}$ dependance of $V_{RMS}$ for the narrow, middle, and deep samples, respectively [57,65].

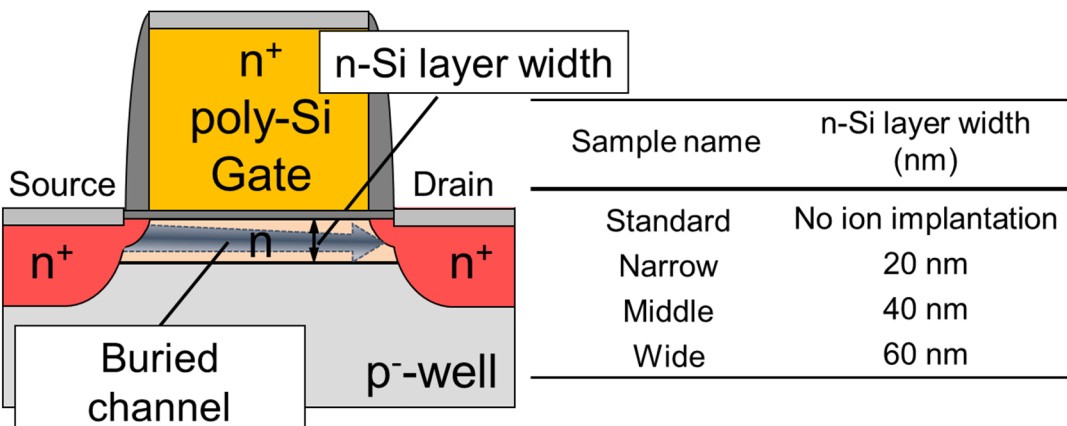

**Figure 23.** Structure of buried channel MOSFETs studied in this work and the channel width for each sample.

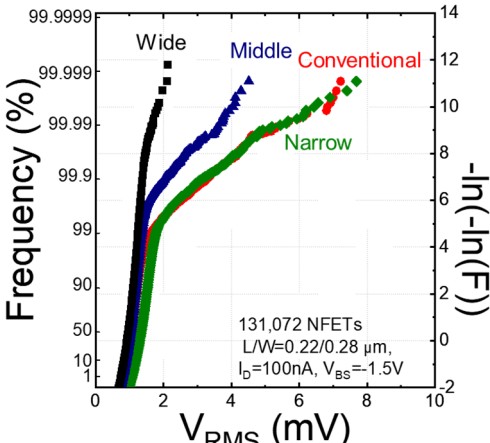

**Figure 24.** Gumbel plots of $V_{RMS}$ for standard, narrow, middle, and deep samples.

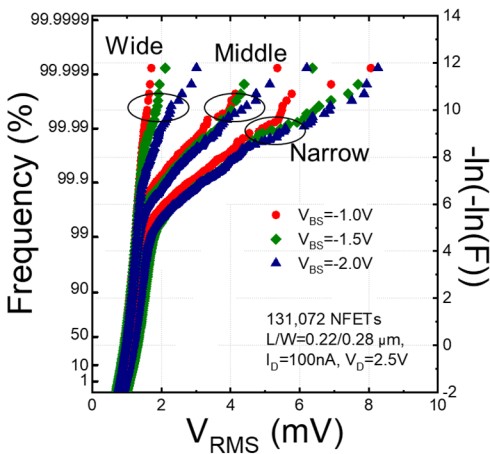

**Figure 25.** $V_{BS}$ dependance of $V_{RMS}$ for narrow, middle, and deep samples.

The channel length and width were 0.22 and 0.28 µm, respectively, $I_{DS}$ was 100 nA, and $V_{BS}$ in Figure 24 was −1.5 V and varied from −1.0 V to −2.0 V in Figure 25, respectively. The $V_{RMS}$ values for the standard and Narrow samples are the same, and the frequency of large $V_{RMS}$ decreases with an increase in n-Si width and/or depth. This means that RTN cannot be decreased by the 20 nm buried channel, but can be decreased by forming a buried channel of 40 nm and more. By increasing the back bias, $V_{RMS}$ increases for all samples, and the effect of $V_{BS}$ remarkably appeared for the wide sample. By applying the back bias,

the channel pushes onto the $SiO_2/Si$ interface, and the channel thickness decreases. The buried channel is extremely effective for decreasing RTN because the channel is separated from $SiO_2/Si$ interface and the wide channel becomes difficult to form the percolation path. Furthermore, the buried channel MOSFET in the isolated well was employed to evaluate $V_{BS}$ dependance [64], and $V_{BS}$ can be varied from 0 V in this structure because the well voltage can be changed freely. The gate length and gate width of the MOSFETs are 0.32 and 0.32 μm, respectively, $I_{DS}$ was 1 μA, vs. was 1.5 V, and $V_{well}$ of the normal well and isolated well were 0 and 1.5 V, respectively; thus, $V_{BS}$ was set at −1.5 and 0 V for the normal well and isolated well, respectively.

Figure 26 shows the Gumbel plot of the $V_{RMS}$ for the buried channel and surface channel MOSFETs with the back bias conditions of 0 and −1.5 V. $V_{RMS}$ of the buried channel MOSFETs at $V_{BS} = −1.5$ V is significantly less than those of surface channel; however, that at $V_{BS} = 0$ V is larger than those of the surface channel even though those of the surface channel do not depend on $V_{BS}$. Figure 27 shows the normal probability plot of the subthreshold swing for the same sample of Figure 26. The subthreshold swing of buried channel MOSFETs with $V_{BS} = 0$ is much smaller than the others. A strong relationship between the subthreshold swing and $V_{RMS}$ has been reported [64]. The result strongly suggests that the increase in $V_{RMS}$ is enlarged by the physical origin, which increases the subthreshold swing, and the origin is an enhancement of the percolation path formation [64]. Note that RTN has to be enhanced by a minimal small gate control effect on the channel. Furthermore, the variability of the threshold voltage is increased using the buried channel MOSFETs; thus, we cannot introduce buried channel MOSFETs when the fixed pattern noise is critical for device performance.

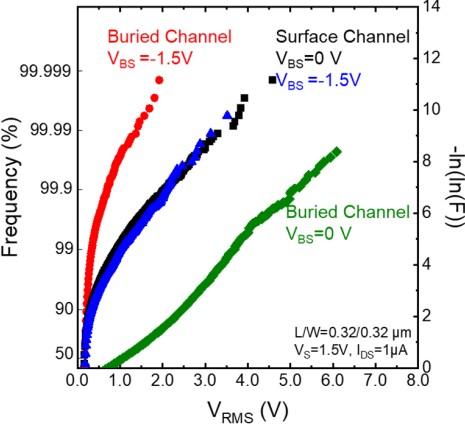

**Figure 26.** Gumbel plot of $V_{RMS}$ for the buried channel and surface channel MOSFETs with the back bias conditions of 0 and −1.5 V.

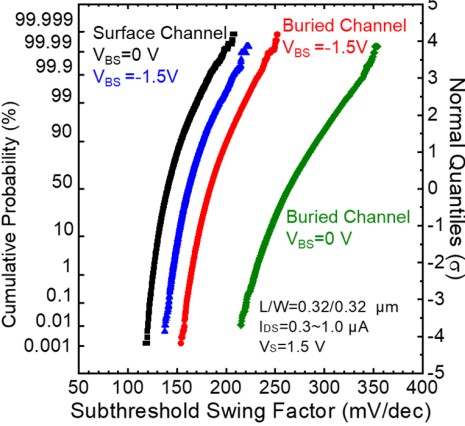

**Figure 27.** $V_{BS}$ dependance of $V_{RMS}$ for the narrow, middle, and deep samples.

### 3.6.2. Asymmetry Source and Drain Width MOSFETs

Figure 28 shows the layout structure of rectangular and trapezoidal shape MOSFETs used in this experiment [62,63]. In the trapezoidal MOSFETs, when current flows from the left to right direction, the source has a wide gate width, and when current flows from the right to left direction, the source has a shallow gate width. The Gumbel plots of $V_{RMS}$ for trapezoidal ((a)$W_D < W_S$ and (b) $W_S < W_D$) and (c) rectangular MOSFETs are shown in Figure 29 [62]. The gate width of rectangular MOSFETs was set as the average of the gate width of trapezoidal MOSFETs. $I_{DS}$ was varied from 0.1 to 11 µA for constant $V_{BS}$ of −1.9 V and $V_{DS}$ of 1.4 V. In (c) rectangular MOSFETs, similar phenomena, as shown in Figure 19, are obtained. In contrast, in the trapezoidal MOSFETs, $V_{RMS}$ increases with an increase in $I_{DS}$ ($V_{GS}$), and those of $W_D < W_S$ are larger than those of $W_S < W_D$. $V_{DS}$ is larger than $V_{GS}$ in this experiment. MOSFETs were operated in the saturation region, and the channel formed near the source. Increasing $I_{DS}$ increases electron density in the channel, and the electron density at the source of MOSFETs with $W_D < W_S$ is less than that with $W_S < W_D$. These characteristics indicate that the influence of a charged trap reduces at a high carrier density condition [39,60,62]. This means that the electron density and the location in the channel are important factors affecting RTN characteristics. RTN characteristics were evaluated for various $V_{DS}$ using rectangular and trapezoidal MOSFETs.

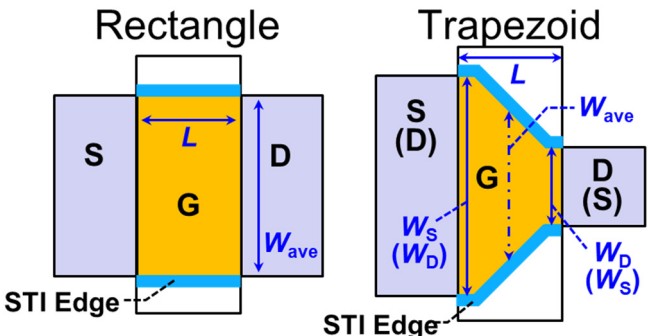

**Figure 28.** Layout diagrams of the measured transistors with various gate shapes.

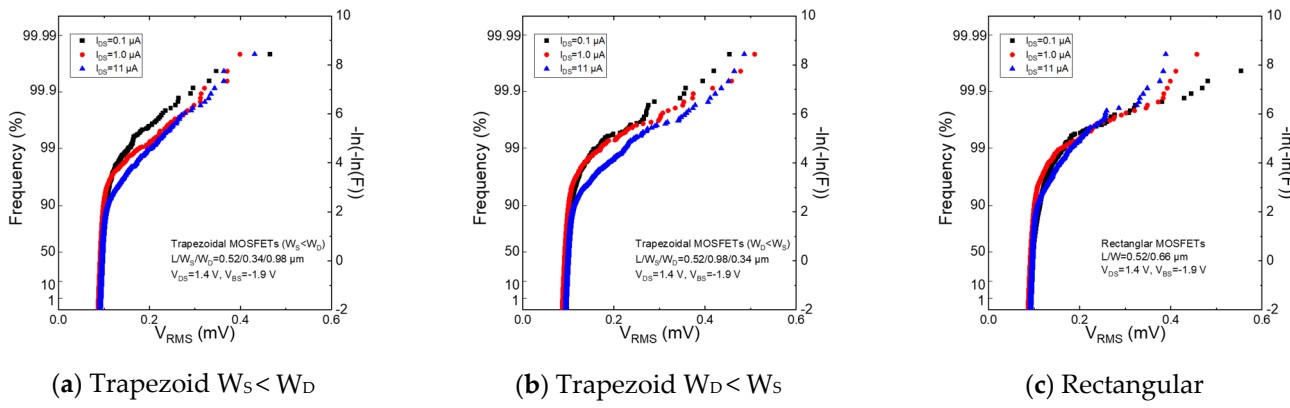

(**a**) Trapezoid $W_S < W_D$  (**b**) Trapezoid $W_D < W_S$  (**c**) Rectangular

**Figure 29.** Gumbel plots of $V_{RMS}$ for trapezoidal and rectangular MOSFETs. $I_{DS}$ was varied from 0.1 to 11 µA for constant $V_{BS}$ of −1.9 V and $V_{DS}$ of 1.4 V.

Figure 30 shows the Gumbel plots of $V_{RMS}$ for trapezoidal ((a) $W_D < W_S$ and (b) $W_S < W_D$) and (c) rectangular MOSFETs. $V_{DS}$ was varied from 0.1 V to 1.4 V for constant $I_{DS}$ of 10 µA and $V_{BS}$ of −1.9 V [63]. The $V_{DS}$ dependance of $V_{RMS}$ for the (c) rectangular and (b) trapezoidal with $W_S < W_D$ MOSFETs is the same, and $V_{RMS}$ increases as $V_{DS}$ increases, monotonically. The dependance of (b) trapezoidal with $W_S < W_D$ on $V_{DS}$ is larger than that of rectangular MOSFETs. In contrast, $V_{RMS}$ increases with an increase in $V_{DS}$ at less than 0.3 V; however, $V_{RMS}$ decreases with an increase in $V_{DS}$ at >0.3V for trapezoidal

MOSFETs with $W_S > W_D$. When $V_{DS}$ is smaller than the pinch-off voltage ($V_{GS}-V_{TH} = 0.3$ V in this experiment), the channel is uniformly formed under the gate oxide, and the channel vanishes at the drain edge at the pinch-off voltage. The vanished region expands with increasing $V_{DS}$. On the other hand, $I_{DS}$ was set at a constant of 10 μA, and this means that $V_{GS}$ decreases as $V_{DS}$ increases at less than a pinch-off voltage of 0.3V. In rectangular MOSFETs, $V_{RMS}$ increases with a decrease in $V_{GS}$, which is the same effect as shown in Figure 7. In trapezoidal MOSFETs with $W_S < W_D$, $V_{DS}$ dependance was enhanced by reducing the channel width. In trapezoidal MOSFETs with $W_D < W_S$, $V_{DS}$ dependance is the same as others at less $V_{DS}$ than a pinch-off voltage of 0.3 V. However, the opposite dependency is obtained at larger $V_{DS}$ than the pinch-off voltage. It is considered that the apparent gate width of MOSFETs ($W_D < W_S$) increases when the pinch-off point reaches the source, and then, the size effect of $V_{RMS}$ shown in Figure 6 is obtained. These data imply that the noise strength depends heavily on operation conditions, which means that the location and electron density in a channel are critical for RTN generation.

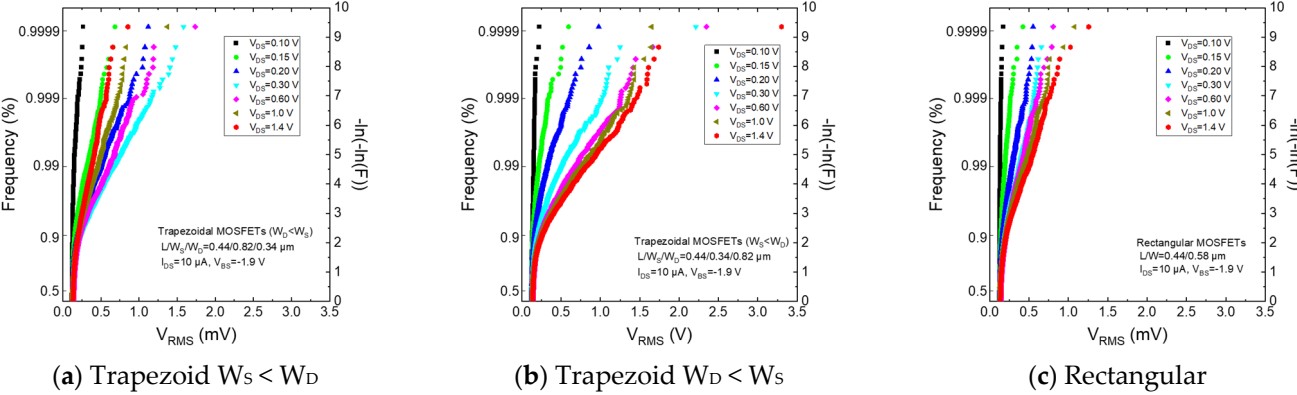

**Figure 30.** Gumbel plots of $V_{RMS}$ for trapezoidal and rectangular MOSFETs. $V_{DS}$ was varied from 0.1 V to 1.4V for constant $I_{DS}$ of 10 μA and $V_{BS}$ of −1.9 V.

Although using trapezoidal MOSFETs in real electronic devices is difficult, changing the shape of MOSFETs is very useful to obtain much information about RTN characteristics. For example, the effect of trap at the isolation edge can be evaluated using octagonal MOSFETs, which have only a gate edge and no shallow trench isolation edge [62].

### 3.6.3. MOSFETs with Atomically Flat Gate Insulator/Si Interface

The roughness of the interface between the gate insulator and Si is essential for MOSFETs. The interface roughness degrades not only electron mobility [66–71] and gate dielectric reliability [72–74], but also noise generation [71,75,76]. An atomically flat interface [77–84] is effective for reducing low-frequency noise [79,83–87].

Figure 31 shows images of an atomically flat surface and as-received Si(100) measured by atomic force microscopy (AFM). The atomically flat surface was formed in the active region with shallow trench isolation and was measured after the gate oxide formation and following oxide stripping [84]. The average roughness (Ra) of the conventional surface is 0.12 nm, which is the same as the initial surface of Si(100). In an atomically flat surface, a step and terrace structure can be obtained, and the step height is the same as the monoatomic step length of Si(100) of 0.135 nm. The terrace width (L) is defined by the following equation using the off-angle (θ) to the just (100) orientation. The average roughness in the trace of the atomically flat interface was less than 0.04 nm, which is the detection limit of our AFM system.

$$L = \frac{0.135}{\tan \theta} \text{ (nm)} \qquad (16)$$

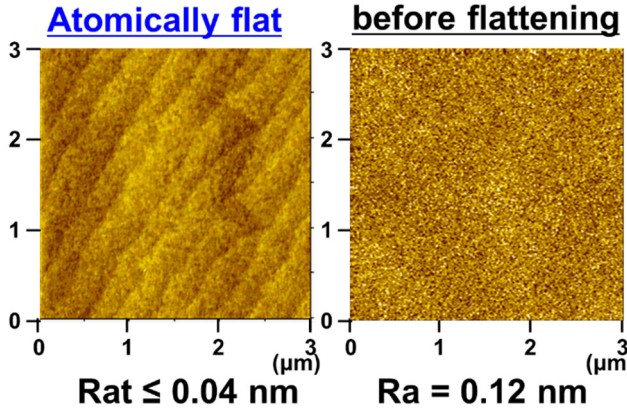

**Figure 31.** Surface images of atomically flat and as-received Si(100) (before flattening) surfaces measured by atomic force microscopy.

Figure 32 shows the Gumbel plot of $V_{RMS}$ for the atomically flat and conventional $SiO_2$/Si interface [84]. The noise of the atomically flat interface is less than that of the conventional interface. This means that introducing the atomically flat interface is extremely effective for reducing RTN as well as 1/f noise [86–88]. The atomically flat surface was formed before gate oxidation in this experiment [84].

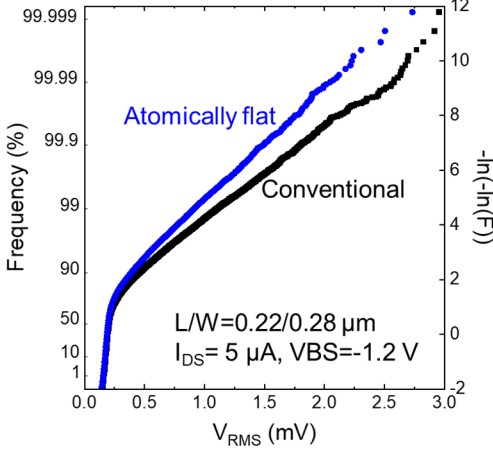

**Figure 32.** Gumbel plot of $V_{RMS}$ for atomically flat and conventional $SiO_2$/Si interface.

To implement the surface flattening process, a low temperature of less than 900 °C and low oxidation species, such as $O_2$ and $H_2O$, must be required [81,82,85]. There is another method for flattening the surface first and keeping it during the process steps preceding gate oxidation [85,87,88]. Other problems exist, such as STI edge shape and dopant segregation, and the solutions to these problems may affect not only MOSFET characteristics, but also noise [84]. The flattening process just before the gate oxidation is superior to the flattening process in the first step for introducing interface flattening between $SiO_2$ and Si, and this can be obtained by low temperature Ar annealing by reducing oxidation species.

## 4. Conclusions

The importance of low-frequency noise in LSI, and various effects on RTN, such as MOSFETs' size, bias and operation conditions, and device structures, are described. The measurement technique using the array test circuit and the extraction of important parameters (time constants and amplitude) in RTN characteristics are also described. Time constants can be extracted essentially using classical equations; however, it is not as simple when downsizing MOSFETs and reducing the number of channel electrons, and the

percolation path is formed. Variability of low-frequency noise increases with shrinkage of MOSFETs. In this paper, we evaluated relatively large planer MOSFETs (L = 0.22~0.4 μm), unfortunately. The size of MOSFETs has been downscaled to less than l0 nm and the structure has changed the planer to FinFET, recently. We have to continue the evaluation of such miniaturized and new structure devices. To assess the effect of this noise on MOSFETs, we have to understand their characteristics statistically, and then, sufficient samples must be accurately evaluated in a short period.

**Funding:** This research received no external funding.

**Acknowledgments:** The author acknowledges Shigetoshi Sugawa, Rihito Kuroda, Tetsuya Goto, Tomoyuki Suwa, Keninchi Abe, Philippe Gaubert, Yuki Kumagai, Takafumi Fujisawa, Hiroyuki Suzuki, Akihiro Yonezawa, Toshiki Obara, Takezo Mawaki, Shinya Ichino, Ryo Akimoto, and Tatsuki Ueta for their measurements and useful discussions and Yutaka Kamata and Katsuhiko Shibusawa for fabrication of the array test circuit and useful discussions.

**Conflicts of Interest:** The authors declare no conflict of interest.

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
