# Peer review of "Evaluation of Low-Frequency Noise in MOSFETs Used as a Key Component in Semiconductor Memory Devices"

_electronics, doi:10.3390/electronics10151759_

Round 1

Reviewer 1 Report

This work presents methods for evaluating low-frequency noise, such as 1/f noise and random telegraph noise, and evaluation results are described. The reviewer found that this work will be of interest to other researchers in scientific and engineering community of MOSFETs. 

Author Response

Thank you very much for kind comment.

Reviewer 2 Report

This paper is well written and the author succinctly and successfully presented  reliability issues (eg., i/f noise and random telegraph noise) for future memory devices. This paper should be accepted in this journal. However I would like to request the author to clearly outline the major differences of this paper with the following published papers in the introduction. 

[1] R. Akimoto et al., "Effect of Drain-to-Source Voltage on Random Telegraph Noise Based on Statistical Analysis of MOSFETs with Various Gate Shapes," 2020 IEEE International Reliability Physics Symposium (IRPS), 2020, pp. 1-6, doi: 10.1109/IRPS45951.2020.9128341.

[2] R Akimoto, R Kuroda, A Teramoto, T Mawaki, S Ichino, T Suwa, S Sugawa "Effect of Drain-to-Source Voltage on Random Telegraph Noise Based on Statistical Analysis:" IEICE Technical Report; IEICE Tech. Rep. 120 (205), 34-39

Author Response

Thank you for the useful comment.

I am a co-author of references [1] and [2], which is the same as ref [63] in this paper.

I refer only [1] because the ref. [2] is written in Japanese.

Then, there is not major differences between this paper and ref [63].

I made the results clearer by increasing data and modifying the graphs for this review paper.

Reviewer 3 Report

See enclosed file.

Author Response

Thank you for very kind reviewing in detail.

Round 2

Reviewer 3 Report

See enclosed file.

Author Response

Thank you for effective indication.

P.2, L47. As you pointed out, I added and changed the sentence, "When P0=P1=1/2 and α=AS/2 are assumed, the LER is given by the following equation from equations (2) and (3)."